# Task-Distributionally Robust Data-Free Meta-Learning

## Abstract

Data-free Meta-learning (DFML) aims to enable efficient learning of new unseen tasks by meta-learning from a collection of pre-trained models without access to their training data. Existing DFML methods construct pseudo tasks from a learnable dataset, which is iteratively inversed from a collection of pre-trained models. However, such distribution of pseudo tasks is not stationary and can be easily corrupted by specific attack, which causes (i) Task-Distribution Shift (TDS): the distribution of tasks will change as the learnable dataset gets updated, making the meta-learner biased and susceptible to overfitting on new tasks, ultimately harming its long-term generalization performance. (ii) Task-Distribution Corruption (TDC): the task distribution can be easily corrupted by deliberately injecting deceptive out-of-distribution models, termed as model poisoning attack. To address these issues, for the first time, we call for and develop robust DFML. Specifically, (i) for handling TDS, we propose a new memory-based DFML baseline (TEAPOT) via meta-learning from a pseudo task distribution. TEAPOT maintains the memory of old tasks to prevent over-reliance on new tasks, with an interpolation mechanism combining classes from different tasks to diversify the pseudo task distribution; (ii) for further defending against TDC, we propose a defense strategy, Robust Model Selection Policy (ROSY), which is compatible with existing DFML methods (e.g., ROSY + TEAPOT). ROSY adaptively ranks and then selects reliable models according to a learnable reliability score, which is optimized by policy gradient due to the non-differentiable property of model selection. Extensive experiments show the superiority of TEAPOT over existing baselines for handling TDS and verify the effectiveness of ROSY + DFML for further improving robustness against TDC.

## 1 Introduction

Data-free Meta-learning (DFML) (Wang et al., 2022; Hu et al., 2023), a newly proposed paradigm of meta-learning (ML), has attracted attention recently thanks to its appealing capability of reusing multiple pre-trained models to obtain a single meta-learner with superior generalization ability in a data-free manner. In contrast, traditional ML methods solve few-shot tasks by meta-learning from a collection of related tasks with available training and testing data. However, in many real scenarios, each task may only have a pre-trained model and the task-specific data is not available after pre-training due to privacy issues. For example, some repositories like GitHub, HuggingFace and Model Zoo provide numerous pre-trained models without training data released. Thus, DFML provides an effective solution to such scenarios, by reusing those pre-trained models in a data-free manner to obtain a meta-learner with superior generalization ability. However, existing DFML methods face vulnerabilities in two critical aspects: **(i) Task-Distribution Shift (TDS)** and **(ii) Task-Distribution Corruption (TDC)**, which have not yet been explored and make it hard to apply in practice.

**TDS arises from the non-stationary distribution of synthetic tasks.** PURER (Hu et al., 2023), the state-of-the-art DFML method, constructs a batch of pseudo tasks at each iteration from a learnable dataset. The learnable dataset is iteratively inversed from a collection of pre-trained models and adversarially optimized with the meta-learner to synthesize pseudo tasks in an increasing difficulty order. However, as the learnable dataset gets updated, the distribution of synthetic tasks will change. Such distribution shift could be large if the learnable dataset and the meta-learner get trained adversarially. This can cause the meta-learner to become biased and prone to overfitting when faced with new tasks, ultimately harming its generalization ability over time. Fig. 1 (a) depicts the dramatic

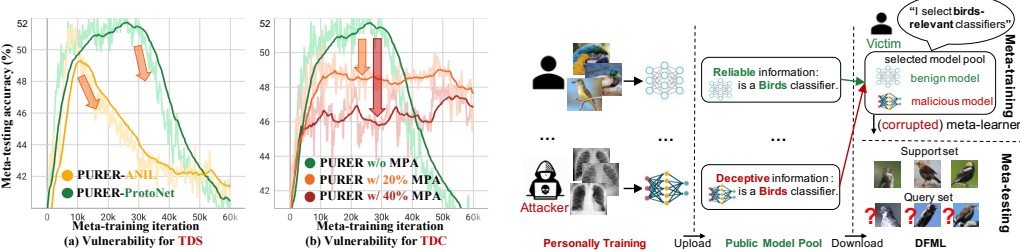

Figure 1: Vulnerabilities for TDS and TDC.                Figure 2: Illustration of model poisoning attack (MPA).

degradation in meta-testing accuracy during the meta-training phase of PURER under the CIFAR-FS 5-way 5-shot setting. Such accuracy degradation is highly undesirable in practical scenarios, making it impractical to snapshot the best meta-learner, particularly when lacking monitoring from the validation set. This highlights the necessity for a robust DFML method that can keep consistently high accuracy over time so that the meta-training phase can be safely terminated after a pre-determined sufficient number of iterations.

**TDC arises when users collect pre-trained models from untrusted sources.** Fig. 2 illustrates this scenario. When employing DFML algorithms to solve rare bird species classification tasks, a user would actively choose relevant models (e.g., bird or animal classifiers) as meta-training resources, according to the domain information attached with the models. PURER(Hu et al., 2023) assume all pre-trained models and their information are reliable. However, this is not always satisfactory. In Fig. 2, an attacker can intentionally release out-of-distribution (OOD) models (e.g., medical image classifiers) while deceptively claiming them as bird classifiers, inducing users to collect them. Attackers can passively release those deceptive models, awaiting collection, or actively send them directly to model collectors with higher privileges. We term this attack model poisoning attack (MPA), injecting malicious models attached with deceptive domain information into the training model pool. We analyze why MPA leads to TDC in Sec. 3.2. Fig. 1 (b) shows the severe accuracy drop of PURER caused by MPA under CIFAR-FS 5-way 5-shot setting. In this case, attackers release malicious models from five OOD datasets, including EuroSAT, ISIC, chestX, Omniglot and MNIST, while claiming them from CIFAR-FS. The severe performance degradation calls for an effective defense strategy, automatically identifying and filtering out those malicious models. It seems straightforward to first evaluate models on some data before leveraging them for meta-training. However, this involves manually collecting model-specific data and then evaluating each model one by one, which may be impractical due to data privacy and the additional cost of data collection and model evaluation. Therefore, we aim to design an elegant strategy that achieves automatic model identification and selection during the training phase with no need for model-specific data and additional cost.

To address these vulnerabilities, for the first time, we call for and develop robust DFML. **(i) For handling TDS**, we propose a new memory-based DFML baseline (TEAPOT) by meta-learning from a pseudo task distribution. TEAPOT preserves the memory of old tasks to prevent over-reliance on specific new tasks, with an interpolation mechanism combining classes from different tasks to further diversify the pseudo task distribution. We emphasize the underlying task distribution should be diversified enough so that the meta-learner can generalize well to unseen tasks. **(ii) For handling TDC**, we propose Robust Model Selection Policy (ROSY), leading to a general defense framework ROSY + DFML. We parameterize ROSY as a learnable weight vector characterizing each model's reliability and thereby enabling the automatic selection of reliable models. Given the non-differentiable nature of model selection, we resort to policy gradient (Williams, 1992) originating from reinforcement learning (RL). The policy is optimized based on the meta-learner's generalization ability on unseen tasks, serving as rewards in the RL framework.

We empirically demonstrate the superiority of our proposed TEAPOT for handling TDS on four datasets (CIFAR-FS, MiniImageNet, VGG-Flower and CUB), outperforming existing baselines by a large margin w.r.t. PEAK accuracy (+ 2.15% ∼ 5.85%) and LAST accuracy (+ 6.09% ∼ 14.45%). We further verify the effectiveness of ROSY + TEAPOT for handling TDC, achieving up to + 2.95% robustness gains across various settings of MPA. We summarize our contributions as four-fold:

- For the first time, we reveal the vulnerabilities of existing DFML methods to the Task-Distribution Shift (TDS) and Task-Distribution Corruption (TDC), highlighting the critical need to develop robust DFML in practice.

- We introduce model poisoning attack (MPA) as a new training-time attack causing TDC, by injecting malicious OOD models with deceptive domain information into the training model pool, which has not been studied yet.

- For handling TDS, we propose a memory-based DFML baseline (TEAPOT) by meta-learning from a pseudo task distribution. For further handling TDC, We propose a defense strategy (ROSY + DFML) to improve the robustness against MPA.

- Experiments on various benchmarks demonstrate the superiority of the proposed TEAPOT for handling TDS and verify the effectiveness and generality of the defense framework ROSY + DFML for further handling TDC.

## 2 RELATED WORK

**Data-free meta-learning (DFML).** DFML aims to enable efficient learning of unseen tasks by meta-learning from a collection of pre-trained models without access to training data. In contrast to traditional data-based meta-learning methods (Yao et al., 2021; Li et al., 2020; Yang et al., 2021; Simon et al., 2022; Ye et al., 2022; Yu et al., 2023a; Jang et al., 2023; Pavasovic et al., 2023; Genewein et al., 2023; Flennerhag et al., 2022; Yao et al., 2022) that rely on large amounts of data, DFML offers a solution to obtain a meta-learner with superior generalization ability from a collection of task-specific pre-trained models with weaker generalization ability. Wang et al. (2022) first introduce the concept of DFML and propose to meta-learn a hyper-network directly outputting meta-initialization in parameter space. More recently, Hu et al. (2023) achieve significant performance improvements by leveraging the underlying data knowledge within each pre-trained model. Their approach, known as PURER, constructs pseudo tasks from an iteratively trained dataset using a collection of pre-trained models, and assumes all those models are reliable. This makes it vulnerable to TDS and TDC discussed in Sec. 1, which poses challenges when applying it in practice. Other works also share a similar spirit of DFML. Kwon et al. (2021) adapt a pre-trained meta-learner to out-of-domain meta-testing tasks in a data-free manner, while we focus on reusing a collection of pre-trained task-specific models. Nava et al. (2023) model the distribution of pre-trained models and directly generate a model for each meta-testing task using additional natural language task description.

**Data-free learning (DFL)** enables the learning process without access to any actual data. This approach is particularly relevant in practical scenarios where data availability is limited due to data privacy, safety, or ethical considerations. Recently, the development of data-free learning has been propelled by techniques such as model inversion (Chen et al., 2019; Mordvintsev et al., 2015; Fang et al., 2022; Zhu et al., 2021; Liu et al., 2021; Truong et al., 2021; Kariyappa et al., 2021; Binici et al., 2022; Liu et al., 2023; Zhang et al., 2021; Wang, 2021; Do et al., 2022; Yu et al., 2023b; Binici et al., 2022), which aims to uncover the underlying data knowledge within pre-trained models. However, existing approaches overlook the non-stationary distribution caused by continuously synthesized data and assume all pre-trained models are reliable, making them susceptible to TDS and TDC.

**Robust meta-learning (RML).** The field of RML has introduced some methods to address the sequential task-distribution shift in meta-learning (Genewein et al., 2023; Setlur et al., 2021). Other methods (Vuorio et al., 2019; Killamsetty et al., 2020; Triantafillou et al., 2020; Yao et al., 2020; Lee et al., 2020; Jeong & Kim, 2020; Jiang et al., 2023) aim to enable effective meta-learning in scenarios with heterogeneous task distributions. However, all these issues and solutions are tailored to data-based meta-learning, while the vulnerabilities associated with data-free meta-learning have not yet been investigated or addressed.

We summarize and compare different settings in Tab. 1, including meta-learning (ML), robust meta-learning (RML), data-free learning (DFL), data-free meta-learning (DFML) and our robust data-free meta-learning (RDFML) in terms of whether they are data-free, consider TDS and TDC

Table 1: Comparisons among different settings.

| Settings | Data-free | TDS | TDC (MPA) | Few-shot | Unseen tasks |
|---|---|---|---|---|---|
| ML | ✗ | ✗ | ✗ | ✓ | ✓ |
| RML | ✗ | ✓ | ✗ | ✓ | ✓ |
| DFL | ✓ | ✗ | ✗ | ✗ | ✗ |
| DFML | ✓ | ✗ | ✗ | ✓ | ✓ |
| **RDFML (ours)** | ✓ | ✓ | ✓ | ✓ | ✓ |

(which arises from MPA), and generalize to few-shot unseen tasks. In summary, our RDFML setting is more comprehensive and practical than existing DFML baslines.

## 3 PROBLEM SETUP

This section begins by introducing the problem of data-free meta-learning (DFML), followed by a description of our proposed attack called model poisoning attack causing TDC.

### 3.1 DATA-FREE META-LEARNING (DFML)

**Meta-training.** We are given a collection of pre-trained models $\mathcal{M}_{pool} = \{M_i\}$, each designed to solve different tasks. The objective of DFML is to meta-learn the meta-learner $\mathcal{A}.[\cdot; \boldsymbol{\theta}_{\mathcal{A}}]$ using $\mathcal{M}_{pool}$, so that $\mathcal{A}[\cdot; \boldsymbol{\theta}_{\mathcal{A}}]$ can be fast adapted to new unseen tasks $\{\mathcal{T}_t^{test}\}$. $t$ is the task index.

**Meta-testing.** We evaluate the meta-learner on 600 unseen $N$-way $K$-shot tasks $\{\mathcal{T}_t^{test} = \{\mathbf{D}_t^{test,s}, \mathbf{D}_t^{test,q}\}\}$. The classes encountered during the meta-testing phase have never been seen during the meta-training phase or in the validation tasks. Each task consists of a support set $\mathbf{D}^{test,s} = (\mathbf{X}^{test,s}, \mathbf{Y}^{test,s})$ with $N$ classes and $K$ instances per class. We utilize the support set $\mathbf{D}^{test,s}$ to adapt the meta learner $\mathcal{A}[\cdot; \boldsymbol{\theta}_{\mathcal{A}}]$ to obtain the task-specific solver $\mathcal{A}[\mathbf{D}^{test,s}; \boldsymbol{\theta}_{\mathcal{A}}]$ and make predictions on its query set $\mathbf{D}^{test,q}$. The overall accuracy is measured by averaging the accuracies across all the meta-testing tasks.

### 3.2 MODEL POISONING ATTACK (MPA)

$\mathcal{M}_{pool}$ denotes the user-collected pre-trained models to train the meta-learner. Generally, each model $M \in \mathcal{M}_{pool}$ released on public repositories is attached with some basic domain information (e.g., the model is an animal classifier). Attackers can execute MPA by injecting OOD models $M_{OOD}$ attached with deceptive domain information (e.g., claiming a Mars-Jupiter classifier as a dog-cat classifier) into $\mathcal{M}_{pool}$. The deceptive domain information can induce users to select those deceptive models.

**Why does MPA lead to TDC?** We can suppose a user wants to collect animal-relevant models as DFML training resources. An attacker releases an OOD model, claiming a Mars-Jupiter classifier as a dog-cat classifier. If the OOD model is wrongly collected, (i) the true "Mars" images inversed from it will be falsely labelled as "dog", contradicting with the true "dog" and other false "dog" images, which thus confuses the meta-learner; (ii) even there is no contradiction, the false "dog" images still cause a huge distribution gap (i.e., the original distribution gap between "Mars" and animal images).

To quantify MPA, we introduce the poisoning rate (PR), which represents the ratio of malicious models present in the training model pool relative to the total number of models.

$$PR = |\mathcal{M}_{OOD}|/|\mathcal{M}_{pool}|. \tag{1}$$

## 4 TASK-DISTRIBUTIONALLY ROBUST DATA-FREE META-LEARNING

In Sec. 4.1, we describe our proposed DFML baseline TEAPOT for handling TDS, followed by two key techniques: pseudo task recovery and interpolated task-memory replay. Moving on to Sec. 4.2, we introduce ROSY + DFML for further handling TDC which arises from MPA.

### 4.1 DATA-FREE META-LEARNING FROM PSEUDO TASK-DISTRIBUTION (TEAPOT)

**Overall objective.** The DFML objective is formulated to meta-train the meta-learner parameterized by $\boldsymbol{\theta}_{\mathcal{A}}$ by minimizing the expected loss with respect to a pseudo task distribution $\hat{p}_{\mathcal{T}}$:

$$\min_{\boldsymbol{\theta}_{\mathcal{A}}} \mathbb{E}_{\hat{\mathcal{T}} \in \hat{p}_{\mathcal{T}}} \mathcal{L}_{task}\left(\hat{\mathbf{D}}^q; \mathcal{A}[\hat{\mathbf{D}}^s; \boldsymbol{\theta}_{\mathcal{A}}]\right), \tag{2}$$

where $\hat{\mathcal{T}} = \{\hat{\mathbf{D}}^s = (\hat{\mathbf{X}}^s, \mathbf{Y}^s), \hat{\mathbf{D}}^q = (\hat{\mathbf{X}}^q, \mathbf{Y}^q)\}$ is the pseudo task sampled from the unknown pseudo task distribution $\hat{p}_{\mathcal{T}}$. $\mathcal{A}[\hat{\mathbf{D}}^s; \boldsymbol{\theta}_{\mathcal{A}}]$ denotes the task-specific adaptation process, i.e., the meta-learner takes the support set $\hat{\mathbf{D}}^s$ as input and outputs the task-specific solver. This can be achieved in several ways, such as in-context (black-box) learning (Brown et al., 2020), gradient optimization (Finn et al., 2017) and non-parametric (metric) learning (Snell et al., 2017). In other words, the original meta-learner is not designed to solve any specific task, while it is shared across $\hat{p}_{\mathcal{T}}$ and can be adapted fast to each specific task drawn from $\hat{p}_{\mathcal{T}}$ via the adaptation process. $\mathcal{L}_{task}(\cdot)$ denotes the

task-level loss function over the query set $\hat{\mathbf{D}}^q$, indicating the effectiveness of the adapted solver. By optimizing this objective, our goal is to train the meta-learner $\mathcal{A}[\cdot; \boldsymbol{\theta}_{\mathcal{A}}]$ to adapt effectively to different tasks drawn from $\hat{p}_{\mathcal{T}}$, resulting in improved generalization and performance on unseen tasks.

**Pseudo task distribution modelling via pseudo task recovery.** Instead of seeking the intractable formula of $\hat{p}_{\mathcal{T}}$, we approximate the task-sampling operation by the pseudo task recovery technique. For each pre-trained model $M$, we aim to recover a pseudo task $\hat{\mathcal{T}} = \{\hat{\mathbf{D}}^s = (\hat{\mathbf{X}}^s, \mathbf{Y}^s), \hat{\mathbf{D}}^q = (\hat{\mathbf{X}}^q, \mathbf{Y}^q)\}$ from $M$ via a randomly initialized generator $G(\cdot; \boldsymbol{\theta}_G)$. The generator $G(\cdot; \boldsymbol{\theta}_G)$ takes the standard Gaussian noise $\mathbf{Z}$ and pre-defined target label $\mathbf{Y}$ as inputs and outputs the recovered data $\hat{\mathbf{X}} = G(\mathbf{Z}, \mathbf{Y}; \boldsymbol{\theta}_{\mathcal{G}})$. We only need to set $\mathbf{Y}$ as a hard label (e.g., $[1, 0, 0]$) or a soft label (e.g., $[0.7, 0.1, 0.2]$), with no need for true class names. To ensure label-conditional generation, we optimize $\boldsymbol{\theta}_G$ by minimizing the classification loss (e.g., cross-entropy (CE)) measuring the classification errors:

$$\min_{\boldsymbol{\theta}_G} \mathcal{L}_{CE}(\hat{\mathbf{X}}, \mathbf{Y}; \boldsymbol{\theta}_G) = \min_{\boldsymbol{\theta}_G} CE(M(\hat{\mathbf{X}}), \mathbf{Y}). \tag{3}$$

We further incorporate a regularization term (Yin et al., 2020) to align the high-level feature map statistics. This term enforces the recovered data $\hat{\mathbf{X}}$ to have similar batch-wise feature-map statistics as those of original training data stored in each BatchNormalization(BN) layer of pre-trained models.

$$\mathcal{L}_{BN}(\hat{\mathbf{X}}; \boldsymbol{\theta}_G) = \sum_l \|\mu_l(\hat{\mathbf{X}}) - \mathrm{BN}_l(\text{running\_mean})\| + \|\sigma_1^2(\hat{\mathbf{X}}) - \mathrm{BN}_1(\text{running\_variance})\|, \tag{4}$$

where $\hat{\mathbf{X}} = G(\mathbf{Z}, \mathbf{Y}; \boldsymbol{\theta}_{\mathcal{G}})$. $\mu_l$ and $\sigma_l^2$ denote the mean and variance of the feature maps from $\hat{\mathbf{X}}$ in the $l^{th}$ convolutional layer. $\mathrm{BN}_l(\text{running\_mean})$ and $\mathrm{BN}_l(\text{running\_variance})$ denote the mean and variance of the feature maps from original training data, which are stored in the $l^{th}$ BN layer.

Integrating above loss functions, we formulate the generation process as the following minimization:

$$\min_{\boldsymbol{\theta}_G} \mathcal{L}_G(\mathbf{Z}, \mathbf{Y}; \boldsymbol{\theta}_G), \quad \text{where} \quad \mathcal{L}_G(\mathbf{Z}, \mathbf{Y}; \boldsymbol{\theta}_G) = \mathcal{L}_{CE} + \mathcal{L}_{BN}. \tag{5}$$

As to recover the pseudo task $\hat{\mathcal{T}}$, we select the recovered data $\hat{\mathbf{X}}$ with the minimum loss value of $\mathcal{L}_G$ during the optimization process of $\boldsymbol{\theta}_G$. Then we randomly split $(\hat{\mathbf{X}}, \mathbf{Y})$ into $\hat{\mathbf{D}}^s = (\mathbf{X}^s, \mathbf{Y}^s)$ and $\hat{\mathbf{D}}^q = (\mathbf{X}^q, \mathbf{Y}^q)$ as the support set and query set, respectively. The task recovery algorithm is summarized in Alg. 3 in App. B.

**Interpolated task-memory replay.** To avoid over-reliance on new tasks, we intuitively introduce a task-memory bank $\mathcal{B}$ to store old recovered tasks and periodically use them to meta-train the meta-learner. However, a fixed number of models with their recovered pseudo tasks are not sufficient to represent the underlying task distribution $\hat{p}_{\mathcal{T}}$, especially under the setting with limited model budgets. We emphasize that the pseudo task distribution should be diverse enough so that the meta-learner can generalize well to unseen tasks. Therefore, we adopt a simple but effective class-wise interpolated task-memory replay technique, randomly selecting $N$ classes along with their corresponding pseudo data from memory tasks, to construct an interpolated $N$-way task representing a mixture of old tasks. It allows the meta-learner to generalize across different tasks and avoid over-reliance on specific tasks. Also, the mechanism for task interpolation can be further explored and improved in future works.

**The choice of $\mathcal{L}_{task}$.** For the proposed TEAPOT, we leverage two forms of the task-level loss function to effectively utilize the supervision from both soft and hard labels. Without loss of generalization, at meta-iteration $k$, the meta learner $\mathcal{A}[\cdot; \boldsymbol{\theta}_{\mathcal{A}}]$ can either learn from the selected models or learn from interpolated tasks from $\mathcal{B}$. For the former, given a batch of $B$ select models $\mathcal{M}_{select} = \{M_i\}$ from $\mathcal{M}_{pool}$, we can obtain a batch of pseudo tasks $\{\hat{\mathcal{T}}_i\}$ from $\mathcal{M}_{select}$ via the pseudo task recovery technique mentioned above. Since $\hat{\mathcal{T}}_i$ and $M_i$ have the same label space, we can leverage the meaningful information contained in the pre-trained models' soft-label predictions. In this case, we realize $\mathcal{L}_{task}$ in Eq. (2) with the KL divergence measuring the difference between the meta-learner's predictions and the soft-label predictions of the pre-trained models. For the latter, we alternatively sample a batch of interpolated tasks $\{\tilde{\mathcal{T}}_i\}$ from $\mathcal{B}$, using the hard labels as the meta-learning supervision. In this case, we realize $\mathcal{L}_{task}$ in Eq. (2) with the cross-entropy loss, comparing the meta-learner's predictions with the hard labels of the interpolated tasks.

$$\boldsymbol{\theta}_{\mathcal{A}}^{(k+1)} \leftarrow \begin{cases} \boldsymbol{\theta}_{\mathcal{A}}^{(k)} - \nabla_{\boldsymbol{\theta}_{\mathcal{A}}^{(k)}} \sum_{i=1}^B KL\left(\mathcal{A}[\hat{\mathbf{D}}_i^s; \boldsymbol{\theta}_{\mathcal{A}}](\hat{\mathbf{X}}_i^q), M_i(\hat{\mathbf{X}}_i^q)\right) & \textit{teacher-guided meta-learning} \\ \boldsymbol{\theta}_{\mathcal{A}}^{(k)} - \nabla_{\boldsymbol{\theta}_{\mathcal{A}}^{(k)}} \sum_{i=1}^B CE\left(\mathcal{A}[\tilde{\mathbf{D}}_i^s; \boldsymbol{\theta}_{\mathcal{A}}](\tilde{\mathbf{X}}_i^q), \mathbf{Y}_i^q\right) & \textit{interpolated task-memory replay.} \end{cases} \tag{6}$$

---

**Algorithm 1:** TEAPOT

---

1 **Input:** Max meta-iterations $N$; the pre-trained model pool $\mathcal{M}_{pool} = \{M_i\}$ and corresponding generators
   $\mathcal{G} = \{G_i(\cdot; \boldsymbol{\theta}_{G_i})\}$; the meta-learner $\mathcal{A}[\cdot; \boldsymbol{\theta}_{\mathcal{A}}^{(1)}]$; the memory bank $\mathcal{B}$.

2 **Output:** Meta-learner $\mathcal{A}[\cdot; \boldsymbol{\theta}_{\mathcal{A}}^{(N)}]$.

3 Randomly initialize the meta-learner $\mathcal{A}(\cdot; \boldsymbol{\theta}_{\mathcal{A}}^{(1)})$, the generators $\{\boldsymbol{\theta}_{G_i}\}$

4 Clear the memory bank $\mathcal{B} \leftarrow [\,]$

5 **for** each meta-iteration $k \leftarrow 1$ **to** $N$ **do**

6      **if not** task-memory replay **then**

7          Randomly select $B$ model(s) $\mathcal{M}_{select}$

8          **for** each teacher $M_i \in \mathcal{M}_{select}$ **do**

             // Pseudo task recovery

9              Generate a pseudo task $\mathcal{T}_i \leftarrow$ RECOVER-TASK-FROM-MODEL($M_i, G_i$) (Alg. 3 in App. B)

10              $\mathcal{B} \leftarrow \mathcal{B} + \mathcal{T}_i$

         // Teacher-guided meta-learning

11          Update meta-learner as $\boldsymbol{\theta}_{\mathcal{A}}^{(k+1)} = \boldsymbol{\theta}_{\mathcal{A}}^{(k)} - \nabla_{\boldsymbol{\theta}_{\mathcal{A}}^{(k)}} \sum_{i=1}^{B} KL\left(\mathcal{A}[\hat{\mathbf{D}}_i^s; \boldsymbol{\theta}_{\mathcal{A}}](\hat{\mathbf{X}}_i^q), M_i(\hat{\mathbf{X}}_i^q)\right)$ (Eq. (6))

12      **else**

         // Interpolated task-memory replay

13          Construct a batch of $B$ interpolated pseudo tasks $\{\tilde{\mathcal{T}}_j\}$ from memory bank $\mathcal{B}$

14          Update meta-learner as $\boldsymbol{\theta}_{\mathcal{A}}^{(k+1)} = \boldsymbol{\theta}_{\mathcal{A}}^{(k)} - \nabla_{\boldsymbol{\theta}_{\mathcal{A}}^{(k)}} \sum_{j=1}^{B} CE\left(\mathcal{A}[\tilde{\mathbf{D}}_j^s; \boldsymbol{\theta}_{\mathcal{A}}](\tilde{\mathbf{X}}_j^q), \mathbf{Y}_j^q\right)$ (Eq. (6))

---

**Flexible choice of the meta-learner $\mathcal{A}$.** Our proposed DFML objective Eq. (2) can extend to optimization-based (e.g., MAML (Finn et al., 2017) or ANIL (Raghu et al., 2019)), as well as metric-based meta-learning methods (e.g., ProtoNet (Snell et al., 2017)). For MAML, the meta-learner $\mathcal{A}_{MAML}$ perform one or few step(s) gradient descent over $\boldsymbol{\theta}_{\mathcal{A}}$ on the support set $\mathbf{X}^s$ to obtain a task-specific solver $F(\cdot)$ parameterized by $\phi$:

$$\mathcal{A}_{MAML}[\mathbf{X}^s; \boldsymbol{\theta}_{\mathcal{A}}](\mathbf{X}^q) = F(\mathbf{X}^q; \boldsymbol{\psi}), \quad \text{s.t.} \quad \boldsymbol{\psi} = \boldsymbol{\theta}_{\mathcal{A}} - \nabla_{\boldsymbol{\theta}_{\mathcal{A}}} CE(F(\mathbf{X}^s; \boldsymbol{\theta}_{\mathcal{A}}), \mathbf{Y}^s) \tag{7}$$

For ProtoNet, the meta learner $\mathcal{A}_{ProtoNet}$ outputs a non-parametric classifier (i.e., nearest centroid classification) via meta-learning a feature extractor $f(\cdot; \boldsymbol{\theta}_{\mathcal{A}})$, modelling the probability of an input $\mathbf{X}^q$ being classified as class $c$ as:

$$[\mathcal{A}_{ProtoNet}[\mathbf{X}^s; \boldsymbol{\theta}_{\mathcal{A}}](\mathbf{X}^q)]_c = \frac{\exp\left(-\|f(\mathbf{X}^q; \boldsymbol{\theta}_{\mathcal{A}}) - \mathbf{C}_c\|\right)}{\sum_{c'} \exp\left(-\|f(\mathbf{X}^q; \boldsymbol{\theta}_{\mathcal{A}}) - \mathbf{C}_{c'}\|\right)}, \tag{8}$$

where $\mathbf{C}_c$ is the average feature embedding calculated with all features in $\mathbf{X}^s$ of class $c$. We summarize the overall algorithm of our proposed DFML baseline TEAPOT in Alg. 1.

## 4.2 ROBUST MODEL SELECTION POLICY (ROSY)

**Policy Modeling.** We parameterize the selection policy as a learnable weight vector $\boldsymbol{W} \in \mathbb{R}^{|\mathcal{M}_{pool}|}$, where $w_i$ characterizes the reliability of $M_i$. At each meta-iteration, we take an action selecting a batch of models $\mathcal{M}_{select}$ from $\mathcal{M}_{pool}$ according to $\boldsymbol{W}$. Here, we use $\pi(\mathcal{M}_{select}|\mathcal{M}_{pool}; \boldsymbol{W})$ to denote the probability of taking this action selecting $\mathcal{M}_{select}$ from $\mathcal{M}_{pool}$:

$$\pi(\mathcal{M}_{select}|\mathcal{M}_{pool}; \boldsymbol{W}) = \prod_{i \in \text{INDEX}(\mathcal{M}_{select})} \left(\frac{e^{w_i}}{\sum_{i'=1}^{|\mathcal{M}_{pool}|} e^{w_{i'}}}\right), \tag{9}$$

where INDEX($\mathcal{M}_{select}$) returns the entry indexes of $\mathcal{M}_{select}$ in $\mathcal{M}_{pool}$. An alternative way to model the selection policy is to adopt a neural network, which can be viewed as further works.

**Defense objective.** We first propose a defense objective to illustrate our goal from a high level: we aim to search the optimal model selection policy (parameterized by $\boldsymbol{W}$) so that the meta-learner (parameterized by $\boldsymbol{\theta}_{\mathcal{A}}^*$) meta-trained with the selected models $\mathcal{M}_{select}$ can generalize well to a handful of disparate validation tasks $\{\mathcal{T}_v^{val} = \{\mathbf{D}_v^{val,s}, \mathbf{D}_v^{val,q}\}\}$. We formulate the defense objective as:

$$\min_{\boldsymbol{W}} \frac{1}{N_v} \sum_{v=1}^{N_v} \mathcal{L}_{task}\left(\mathbf{D}_v^{val,q}; \mathcal{A}[\mathbf{D}_v^{val,s}; \boldsymbol{\theta}_{\mathcal{A}}^*(\boldsymbol{W})]\right), \text{where} \quad \boldsymbol{\theta}_{\mathcal{A}}^* = \text{DFML}(\mathcal{M}_{select}; \boldsymbol{W}), \tag{10}$$

where DFML($\mathcal{M}_{select}$) return the meta-learner trained with $\mathcal{M}_{select}$ via certain DFML algorithm.

**Bi-level Optimization via RL.** The sampling operation (i.e., $\mathcal{M}_{select} \leftarrow \mathcal{M}_{pool}$) is non-differentiable, making the optimization Eq. (10) intractable. Therefore, we adopt a policy gradient method REINFORCE (Williams, 1992) to reformulate Eq. (10) to an differentiable form Eq. (11). Specifically, at meta-iteration $k$, we regard the average accuracy on $N_v$ validation tasks as the reward $\sum_{v=1}^{N_v} R_v^{(k)}$. Intuitively, if the action $\mathcal{M}_{select}^{(k)} \leftarrow \mathcal{M}_{pool}$ leads to an increasing reward, we will optimize the policy so that the probability

---

**Algorithm 2:** ROSY + DFML

**1 Input:** Max meta-iterations $N$; the pre-trained model pool $M_{pool}$; DFML($\cdot$) algorithm; the validation tasks $\{\mathcal{T}_v^{val}\}$.

**2** Initialize the learnable weights $\boldsymbol{W}^{(1)}$

**3 for** each meta-iteration $k \leftarrow 1$ **to** $N$ **do**

**4**     Calculate the probability $\boldsymbol{P}^{(k)} \leftarrow$ SOFTMAX($\boldsymbol{W}^{(k)}$)

**5**     Select $B$ models as $\mathcal{M}_{select}^{(k)}$ via the probability $\boldsymbol{P}^{(k)}$

**6**     Obtain the updated meta-learner $\boldsymbol{\theta}_{\mathcal{A}}^{(k+1)}$ via one-iteration DFML algorithm with selected $\mathcal{M}_{select}^{(k)}$

**7**     Calculate the reward $\frac{1}{N_v} \sum_{v=1}^{N_v} R_v^{(k)}$

**8**     Optimize the policy to $\boldsymbol{W}^{(k+1)}$ via Eq. (11)

---

of taking this action $\pi(\mathcal{M}_{select}^{(k)}|\mathcal{M}_{pool})$ will increase, and vice versa. To reduce the gradient variance, we introduce the baseline function $b$ as the moving average of all past rewards.

$$\boldsymbol{W}^{(k+1)} \leftarrow \boldsymbol{W}^{(k)} + \nabla_{\boldsymbol{W}^{(k)}} \left[ \log \pi(\mathcal{M}_{select}^{(k)}|\mathcal{M}_{pool}; \boldsymbol{W}^{(k)}) \times \left( \frac{1}{N_v} \sum_{v=1}^{N_v} R_v^{(k)} - b \right) \right]. \quad (11)$$

We summarize the overall algorithm of ROSY + DFML in Alg. 2.

## 5 EXPERIMENTS

In Sec. 5.1, we empirically demonstrate the superiority of TEAPOT over existing baselines for handling TDS and verify the effectiveness of ROSY + DFML for further handling TDC under MPA.

**Experimental setup.** We adopt CIFAR-FS, MiniImageNet, VGG-Flower and CUB datasets, commonly used in recent meta-learning works (Yao et al., 2021; Tack et al., 2022). Following standard splits (Triantafillou et al., 2020), we split each dataset into the meta-training, meta-validating and meta-testing subsets with disjoint label spaces. Following Wang et al. (2022); Hu et al. (2023), we collect 100 models pre-trained on 100 $N$-way tasks sampled from the meta-training subset and those models are used as the meta-training resources. **For model architecture**, we adopt Conv4 as the architecture of the meta-learner and the pre-trained models for a fair comparison with existing works. We provide the detailed structure for the generator in App. C. **For hyperparameters**, we implement TEAPOT-ANIL with an inner learning rate of 0.01 and an outer learning rate of 0.001. We implement TEAPOT-ProtoNet with a learning rate of 0.001. We set the budget of the memory bank as 20 tasks. We report the average accuracy over 600 meta-testing tasks. We leave the other setup in App. A.

### 5.1 DATA-FREE META-LEARNING W/O ATTACK

**Baselines. (i) RANDOM.** Learn a classifier using the support set from scratch for each meta-testing task. **(ii) AVERAGE.** Average all pre-trained models and then finetune it using the support set. **(iii) DRO** (Wang et al., 2022). Meta-learn a hyper-network to fuse all pre-trained models into one single model, which serves as the meta-initialization and can be adapted to each meta-testing task using the support set. **(iv) PURER-[·]** (Hu et al., 2023). Adversarially train the meta-learner with a learnable dataset, where a batch of pseudo tasks is sampled for meta-training at each iteration. [·] indicates the meta-learning algorithm, such as ANIL and ProtoNet.

**Metrics. (i) PEAK** denotes the peak meta-testing accuracy obtained by the checkpoints with the highest meta-validating accuracy. **(ii) LAST** denotes the meta-testing accuracy obtained by the checkpoints in the last iteration. **(iii) VARIATION** denotes the value of "LAST - PEAK", indicating the variation of meta-testing accuracy through the whole meta-training phase.

**Main results.** Tab. 2 shows the results for 5-way classification compared with existing baselines. We list our main findings as follows: **(i)** TEAPOT achieve significantly **higher PEAK accuracy** on all four datasets. Compared with the best baseline, TEAPOT achieves 2.15% $\sim$ 5.85% performance gains for 1-shot learning and 2.63% $\sim$ 4.44% performance gains for 5-shot learning w.r.t. the PEAK

Table 2: Compare to existing baselines w/o model poisoning attack. † denotes our proposed method.

| Method | CIFAR-FS (Bertinetto et al., 2018) | | | | MiniImageNet (Vinyals et al., 2016) | | | |
|---|---|---|---|---|---|---|---|---|
| | 5-way 1-shot | | 5-way 5-shot | | 5-way 1-shot | | 5-way 5-shot | |
| | PEAK | LAST | PEAK | LAST | PEAK | LAST | PEAK | LAST |
| RANDOM | 28.59 ± 0.56 | 28.59 ± 0.56 | 34.77 ± 0.62 | 34.77 ± 0.62 | 25.06 ± 0.50 | 25.06 ± 0.50 | 28.10 ± 0.52 | 28.10 ± 0.52 |
| AVERAGE | 23.96 ± 0.53 | 23.96 ± 0.53 | 27.04 ± 0.51 | 27.04 ± 0.51 | 23.79 ± 0.48 | 23.79 ± 0.48 | 27.49 ± 0.50 | 27.49 ± 0.50 |
| DRO | 30.43 ± 0.43 | 29.35 ± 0.41 | 36.21 ± 0.51 | 35.28 ± 0.49 | 27.56 ± 0.48 | 25.22 ± 0.42 | 30.19 ± 0.43 | 28.43 ± 0.44 |
| PURER-ANIL | 35.31 ± 0.70 | 26.40 ± 0.43 | 51.63 ± 0.78 | 41.24 ± 0.68 | 30.20 ± 0.61 | 23.05 ± 0.36 | 40.78 ± 0.62 | 29.60 ± 0.53 |
| PURER-ProtoNet | 36.26 ± 0.62 | 27.01 ± 0.58 | 52.67 ± 0.68 | 40.53 ± 0.67 | 30.46 ± 0.64 | 24.00 ± 0.52 | 41.00 ± 0.58 | 31.32 ± 0.52 |
| TEAPOT-ANIL† | 40.39 ± 0.79 | 39.69 ± 0.79 | 55.31 ± 0.75 | 52.92 ± 0.75 | 32.58 ± 0.68 | 29.76 ± 0.61 | **43.63 ± 0.72** | **42.45 ± 0.67** |
| TEAPOT-ProtoNet† | **40.80 ± 0.78** | **40.28 ± 0.79** | **57.11 ± 0.78** | **55.69 ± 0.76** | **32.61 ± 0.64** | **31.97 ± 0.61** | 42.93 ± 0.65 | 41.28 ± 0.64 |

| Method | VGG-Flower (Nilsback & Zisserman, 2008) | | | | CUB (Wah et al., 2011) | | | |
|---|---|---|---|---|---|---|---|---|
| | 5-way 1-shot | | 5-way 5-shot | | 5-way 1-shot | | 5-way 5-shot | |
| | PEAK | LAST | PEAK | LAST | PEAK | LAST | PEAK | LAST |
| RANDOM | 38.39 ± 0.71 | 38.39 ± 0.71 | 48.18 ± 0.65 | 48.18 ± 0.65 | 26.26 ± 0.48 | 26.26 ± 0.48 | 29.89 ± 0.55 | 29.89 ± 0.55 |
| AVERAGE | 24.52 ± 0.46 | 24.52 ± 0.46 | 32.78 ± 0.53 | 32.78 ± 0.53 | 24.53 ± 0.46 | 24.53 ± 0.46 | 28.00 ± 0.47 | 28.00 ± 0.47 |
| DRO | 40.02 ± 0.72 | 38.98 ± 0.74 | 50.22 ± 0.68 | 49.13 ± 0.70 | 28.33 ± 0.69 | 26.01 ± 0.68 | 31.24 ± 0.76 | 29.39 ± 0.70 |
| PURER-ANIL | 51.34 ± 0.80 | 45.02 ± 0.68 | 67.26 ± 0.75 | 62.54 ± 0.72 | 31.29 ± 0.64 | 25.05 ± 0.62 | 43.34 ± 0.59 | 32.08 ± 0.60 |
| PURER-ProtoNet | 53.90 ± 0.76 | 47.12 ± 0.71 | 68.01 ± 0.68 | 64.51 ± 0.67 | 31.62 ± 0.63 | 27.23 ± 0.61 | 45.36 ± 0.71 | 35.32 ± 0.66 |
| TEAPOT-ANIL† | 55.28 ± 0.79 | 54.86 ± 0.76 | 69.03 ± 0.78 | 68.52 ± 0.75 | 35.65 ± 0.72 | 34.32 ± 0.69 | 47.24 ± 0.72 | 46.28 ± 0.65 |
| TEAPOT-ProtoNet† | **57.31 ± 0.85** | **56.79 ± 0.80** | **71.12 ± 0.71** | **70.60 ± 0.69** | **37.47 ± 0.73** | **36.67 ± 0.71** | **48.68 ± 0.71** | **47.64 ± 0.68** |

Figure 3: Performance stability of TEAPOT compared with PURER during the whole meta-training phase (60k meta-iterations). Our TEAPOT can achieve significantly higher and more stable meta-testing accuracy.

accuracy. **(ii)** TEAPOT achieve significantly **higher LAST accuracy and less VARIATION** on all four datasets. Compared with the best baseline, TEAPOT achieves 6.75% ∼ 10.93% performance gains for 1-shot learning and 6.09% ∼ 14.45% performance gains for 5-shot learning w.r.t. the LAST accuracy. More generally, as we can see in Fig. 3, TEAPOT can maintain stable and high meta-testing accuracy during the whole meta-training phase (60k meta-iterations), while PURER suffers from dramatic performance degradation. The significant performance drop of PURER results from over-reliance on new tasks, making it biased and prone to overfitting, thus harming its generalization ability over time. **(iii) Simply AVERAGE all models, surprisingly, even performs worse than RANDOM.** This is because each model is trained to solve different tasks, thus lacking precise correspondence in parameter space. In Tab. 9 of App. D, we dig into this phenomenon and reveal that AVERAGE is very sensitive to the number of pre-trained models.

Table 3: Ablation studies.

| Method | 5-way 1-shot | | 5-way 5-shot | |
|---|---|---|---|---|
| | PEAK | LAST | PEAK | LAST |
| V | 35.34 ± 0.68 | 26.76 ± 0.64 | 50.31 ± 0.72 | 39.22 ± 0.74 |
| V + M | 36.02 ± 0.70 | 28.33 ± 0.68 | 51.23 ± 0.72 | 41.54 ± 0.70 |
| V + M + I | 38.76 ± 0.74 | 38.42 ± 0.72 | 54.91 ± 0.74 | 53.52 ± 0.74 |
| V + M +I + T | 40.80 ± 0.78 | 40.28 ± 0.79 | 57.11 ± 0.78 | 55.69 ± 0.76 |

**Ablation studies.** Tab. 3 analyzes the effectiveness of each component on CIFAR-FS. We first introduce the **V** (Vanilla) meta-learning from pseudo tasks synthesized from models without the memory bank. V suffers from TDS, leading to a dramatic performance drop. Simply adding a Memory bank (**M**) does not work. This is because the task distribution is not diversified enough to enable the meta-learner to generalize to unseen tasks. By adding **I** (Interpolated task-memory replay), it achieves much high PEAK and less VARIATION, indicating the interpolation mechanism does help improve robustness against TDS and enhance generalization ability. We also observe an improvement by leveraging the soft-label supervision from pre-trained Teacher models (i.e., **T**). With all components, we achieve the best with a boosting improvement of PEAK and much less VARIATION, demonstrating the effectiveness of the joint schema. Tab. 6 of App. D shows increasing the memory bank size can result in improved meta-learning performance.

## 5.2 DATA-FREE META-LEARNING W/ ATTACK

**MPA setup.** To simulate MPA, we randomly inject a certain number of OOD models per-trained from EuroSAT (Helber et al., 2019), ISIC (Tschandl et al., 2018; Codella et al., 2019), chestX (Wang

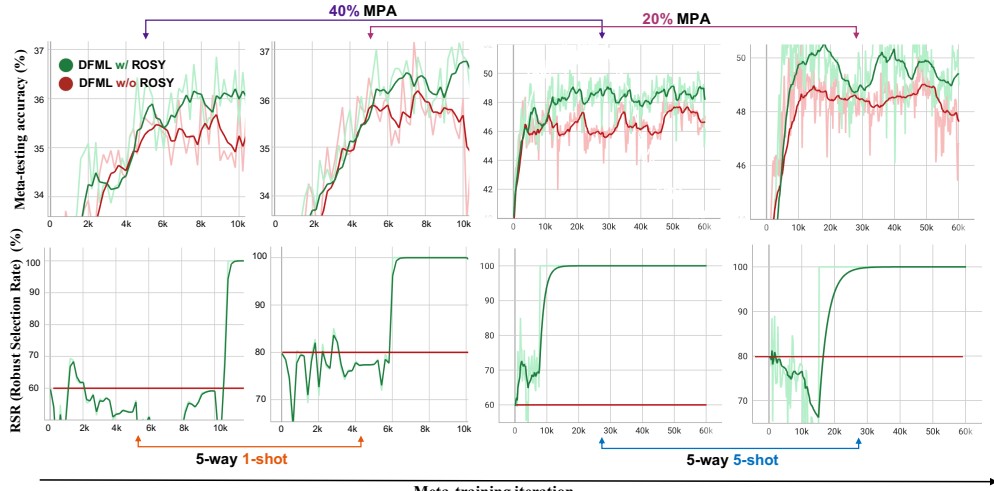

Figure 4: (Top) Performance gains brought by ROSY on CIFAR-FS. (Bottom) Trend of RSR.

et al., 2017), Omniglot (Lake et al., 2015) and MNIST (LeCun et al., 2010) into the training model pool. The other 100 benign models are pre-trained on CIFAR-FS. Injecting those OOD models can lead to severe TDC, and the reasons are fully illustrated in Sec. 3.2.

**Results.** Tab. 4 shows the effectiveness of ROSY on CIFAR-FS under different poisoning rates. We summarize the main findings: (i) ROSY can obtain consistent defense gains (1.12% $\sim$ 2.59% for 1-shot learning and 1.56% $\sim$ 2.95% for 5-shot learning) under poisoning rates ranging from 10% to 80%, indicating the effectiveness of ROSY for handling TDC that arises from model poisoning attack. (ii) With the increase of the noise ratio, ROSY achieves more significant improvements, sug-

Table 4: ROSY against MPA of different poisoning rates.

| Poisoning Rate | Method | 5-way 1-shot | 5-way 5-shot |
|---|---|---|---|
| 10% | TEAPOT | $34.37 \pm 0.71$ | $47.51 \pm 0.72$ |
| | TEAPOT + ROSY | $35.57 \pm 0.67_{+1.20\%}$ | $49.34 \pm 0.72_{+1.83\%}$ |
| 20% | TEAPOT | $33.12 \pm 0.69$ | $46.79 \pm 0.72$ |
| | TEAPOT + ROSY | $34.24 \pm 0.69_{+1.12\%}$ | $48.72 \pm 0.74_{+1.93\%}$ |
| 40% | TEAPOT | $33.03 \pm 0.71$ | $45.25 \pm 0.74$ |
| | TEAPOT + ROSY | $35.22 \pm 0.75_{+2.19\%}$ | $46.81 \pm 0.74_{+1.56\%}$ |
| 60% | TEAPOT | $31.20 \pm 0.68$ | $42.65 \pm 0.72$ |
| | TEAPOT + ROSY | $33.17 \pm 0.66_{+1.97\%}$ | $45.60 \pm 0.69_{+2.95\%}$ |
| 80% | TEAPOT | $30.47 \pm 0.64$ | $40.96 \pm 0.70$ |
| | TEAPOT + ROSY | $33.06 \pm 0.68_{+2.59\%}$ | $43.58 \pm 0.65_{+2.62\%}$ |

gesting that involving the automatic model selection does improve the robustness of DFML. In Tab. 7 of App. D, we also verify the effectiveness of ROSY on other DFML algorithms like PURER, which means ROSY + DFML is a general framework for existing DFML algorithms.

**Analysis of learnable weights.** To figure out how ROSY works, we take a deep look at the trend of those learnable weights $W$ (see Eq. (11)) during the meta-training phase. We first introduce an indicator named RSR (Robust Sampling Rate): $RSR = \sum_{i \in \text{INDEX}(\mathcal{M}_{benign})} \left( \frac{e^{w_i}}{\sum_{i'=1}^{|\mathcal{M}_{pool}|} e^{w_{i'}}} \right)$, where $\text{INDEX}(\mathcal{M}_{benign})$ returns the entry indexes of all benign models $\mathcal{M}_{benign}$ in $\mathcal{M}_{pool}$. RSR indicates the probability of selecting benign models instead of those OOD models with deceptive model information. The increasing value of RSR in Fig. 4 (Bottom) shows that ROSY gradually learns how to identify OOD models and tends to select the benign models with an increasing probability.

## 6 CONCLUSION

In this work, for the first time, we reveal the vulnerabilities of DFML to Task-Distribution Shift (TDS) and Task-Distribution Corruption (TDC) that arises from model poisoning attack (MPA). For handling TDS, we propose a memory-based baseline TEAPOT by meta-learning from a pseudo task distribution. For further handling TDC, we propose a general defense framework ROSY + DFML to automatically filter out deceptive models via reinforcement learning. Our experiments demonstrate the superiority of TEAPOT for handling TDS and verify the effectiveness of ROSY + DFML for further alleviating TDC. Future work includes extending the proposed methods to more complex cases where pre-trained models are black-box.

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

# Appendix

## A    ADDITIONAL EXPERIMENTAL SETUP

Following Wang et al. (2022); Hu et al. (2023), we adopt Conv4 as the architecture of the meta-learner and the pre-trained models for a fair comparison, which is commonly used in recent meta-learning works. Conv4 consists of four convolutional blocks. Each block consists of 32 $3 \times 3$ filters, a BatchNorm, a ReLU and a $2 \times 2$ max-pooling. We implement TEAPOT-ANIL (Raghu et al., 2019) with an inner learning rate of 0.01 and an outer learning rate of 0.001. We implement TEAPOT-ProtoNet (Snell et al., 2017) with a learning rate of 0.001. We set the budget of the memory bank as 20 tasks. Following Wang et al. (2022); Hu et al. (2023), we collect 100 models pre-trained on 100 $N$-way tasks sampled from the meta-training subset. Those pre-trained models are trained using an Adam optimizer with a learning rate of 0.01 for 60 epochs. For generator training, we adopt Adam optimizer with a learning rate of 0.001 for 200 epochs. We provide the architecture of the generator in App. C. We switch from task-memory replay to generating new tasks if the meta-validating accuracy has not increased for 10 meta-iterations. All experiments are conducted on the NVIDIA GeForce RTX 3090 GPU.

## B    ADDITIONAL ALGORITHM

---

**Algorithm 3:** Recover-Task-From-Model Subroutine

---

1  **Input:** Max generation-iterations $N_G$; the pre-trained model $M$ and the corresponding generator $G(\cdot; \boldsymbol{\theta}_G)$.
2  **Function** `Recover-Task-From-Model(M, G)`:
3       Initialize $minLoss \leftarrow +\infty$, $bestTask \leftarrow None$
4       Randomly initialize $\boldsymbol{\theta}_G$
5       Pre-define the labels $\mathbf{Y}$
6       **for** each iteration $n_g \leftarrow 1$ **to** $N_G$ **do**
7           Randomly sample a batch of noise $\mathbf{Z}$ from the standard Gaussian distribution
8           $\mathbf{X}^{(n_g)} \leftarrow G(\mathbf{Z}, \mathbf{Y}; \boldsymbol{\theta}_G^{(n_g)})$
9           Calculate the loss value $L_G^{(n_g)} \leftarrow \mathcal{L}_G(\mathbf{Z}, \mathbf{Y}; \boldsymbol{\theta}_G^{(n_g)})$
10          $\boldsymbol{\theta}_G^{(n_g+1)} \leftarrow \boldsymbol{\theta}_G^{(n_g)} - \nabla_{\boldsymbol{\theta}_G} \mathcal{L}_G^{(n_g)}$ (Eq. (5))
11          **if** $\mathcal{L}_G^{(n_g)} < minLoss$ **then**
12              $minLoss \leftarrow \mathcal{L}_G^{(n_g)}$
13              $bestTask \leftarrow \mathbf{X}^{(n_g)}$
14      Randomly split $bestTask$ into the support set $\hat{\mathbf{D}}^s$ and the query set $\hat{\mathbf{D}}^q$
15      **return** $bestTask$ with $\hat{\mathbf{D}}^s$ and $\hat{\mathbf{D}}^q$

---

## C    ARCHITECTURE OF CONDITIONAL GENERATOR

Tab. 5 lists the structure of the conditional generator in our proposed TEAPOT. The generator takes the standard Gaussian noise and the one-hot label embedding as inputs and outputs the recovered data. Here, $d_{\boldsymbol{z}}$ is the dimension of Gaussian noise data $\boldsymbol{z}$, which is set as 256 in practice. The $negative\_slope$ of LeakyReLU is 0.2. We set $img\_size$ as 32. We set the number of channels $nc$ as 3 for color image recovery and the number of convolutional filters $nf$ as 64.

## D    ADDITIONAL EXPERIMENTS

**Effect of the memory budgets.** Tab. 6 shows the effect of the memory-bank size. We conduct experiments on CIFAR-FS for 5-way 1-shot and 5-shot learning. We measure the size with one $N$-way $K$-shot task. Within a specific range, increasing the memory-bank size resulted in improved generalization ability. Intuitively, a larger memory bank has the capacity to store a greater number

Table 5: Detailed structure of conditional generator. We highlight the dimension change in red.

| Notion | Description |
|---|---|
| $img\_size \times img\_size$ | resolution of recovered image |
| $bs$ | batch size |
| $nc$ | number of channels of recovered image |
| $nf$ | number of convolutional filters |
| FC($\cdot$) | fully connected layer; |
| BN($\cdot$) | batch normalization layer |
| Conv2D($input, output, filter\_size, stride, padding$) | convolutional layer |

| Structure | Dimension | |
|---|---|---|
| | Before | After |
| $z \in \mathbb{R}_{d_z} \sim \mathcal{N}(0, 1)$ | — | $[\, bs, d_z \,]$ |
| $y \in \mathbb{R}_{d_y}$ | — | $[\, bs, d_y \,]$ |
| FC$_1(z)$ | $[\, bs, d_z \,]$ | $[\, bs, nf \times (img\_size//4) \times (img\_size//4) \,]$ |
| FC$_2(y)$ | $[\, bs, d_y \,]$ | $[\, bs, nf \times (img\_size//4) \times (img\_size//4) \,]$ |
| Concatenate(FC$_1(z)$,FC$_2(y)$) | — | $[\, bs, 2 \times nf \times (img\_size//4) \times (img\_size//4) \,]$ |
| Reshape | $[\, bs, 2 \times nf \times (img\_size//4) \times (img\_size//4) \,]$ | $[\, bs, 2 \times nf, (img\_size//4), (img\_size//4) \,]$ |
| BN | $[\, bs, 2 \times nf, (img\_size//4), (img\_size//4) \,]$ | $[\, bs, 2 \times nf, (img\_size//4), (img\_size//4) \,]$ |
| Upsampling | $[\, bs, 2 \times nf, (img\_size//4), (img\_size//4)) \,]$ | $[\, bs, 2 \times nf, (img\_size//2), (img\_size//2)) \,]$ |
| Conv2D($2 \times nf, \ 2 \times nf, \ 3, \ 1, \ 1$) | $[\, bs, 2 \times nf, (img\_size//2), (img\_size//2)) \,]$ | $[\, bs, 2 \times nf, (img\_size//2), (img\_size//2)) \,]$ |
| BN, LeakyReLU | $[\, bs, 2 \times nf, (img\_size//2), (img\_size//2)) \,]$ | $[\, bs, 2 \times nf, (img\_size//2), (img\_size//2)) \,]$ |
| Upsampling | $[\, bs, 2 \times nf, (img\_size//2), (img\_size//2)) \,]$ | $[\, bs, 2 \times nf, img\_size, img\_size \,]$ |
| Conv2D($2 \times nf, \ nf, \ 3, \ 1, \ 1$) | $[\, bs, 2 \times nf, img\_size, img\_size \,]$ | $[\, bs, nf, img\_size, img\_size \,]$ |
| BN, LeakyReLU | $[\, bs, nf, img\_size, img\_size \,]$ | $[\, bs, nf, img\_size, img\_size \,]$ |
| Conv2D($nf, \ nc, \ 3, \ 1, \ 1$) | $[\, bs, nf, img\_size, img\_size \,]$ | $[\, bs, nc, img\_size, img\_size \,]$ |
| Sigmoid | $[\, bs, nc, img\_size, img\_size \,]$ | $[\, bs, nc, img\_size, img\_size \,]$ |

of old synthetic tasks, thereby preventing the meta-learner from over-reliance on specific tasks and leading to enhanced generalization ability.

Table 6: Effect of memory-bank size.

| Size | 5-way 1-shot | 5-way 5-shot |
|---|---|---|
| 1 | $34.06 \pm 0.66$ | $48.13 \pm 0.76$ |
| 10 | $39.26 \pm 0.73$ | $53.99 \pm 0.76$ |
| 20 | $\mathbf{40.80 \pm 0.78}$ | $\mathbf{57.11 \pm 0.78}$ |

Table 7: Effect of ROSY + PURER for handling TDC.

| Poisoning Rate | Method | 5-way 1-shot | 5-way 5-shot |
|---|---|---|---|
| 20% | PURER | $35.52 \pm 0.55$ | $50.16 \pm 0.64$ |
| | PURER + ROSY | $37.00 \pm 0.62_{+1.48\%}$ | $52.18 \pm 0.65_{+2.02\%}$ |
| 40% | PURER | $35.37 \pm 0.64$ | $48.39 \pm 0.63$ |
| | PURER + ROSY | $36.92 \pm 0.55_{+1.55\%}$ | $50.25 \pm 0.66_{+1.86\%}$ |

Table 8: Effect of number of pre-trained models on TEAPOT.

| Number | 5-way 1-shot | 5-way 5-shot |
|---|---|---|
| 10 | $38.32 \pm 0.75$ | $49.83 \pm 0.74$ |
| 50 | $39.15 \pm 0.75$ | $52.41 \pm 0.76$ |
| 100 | $\mathbf{40.80 \pm 0.78}$ | $\mathbf{57.11 \pm 0.78}$ |

Table 9: Effect of number of pre-trained models on AVERAGE.

| Number | 5-way 1-shot | 5-way 5-shot |
|---|---|---|
| 0 (RANDOM) | $\mathbf{28.59 \pm 0.56}$ | $34.77 \pm 0.62$ |
| AVERAGE 10 | $27.99 \pm 0.59$ | $\mathbf{36.92 \pm 0.67}$ |
| AVERAGE 50 | $24.05 \pm 0.51$ | $28.16 \pm 050$ |
| AVERAGE 100 | $23.96 \pm 0.53$ | $27.04 \pm 0.51$ |

**Effect of ROSY on other DFML algorithms.** To verify the great generality of SORY, we apply it to PURER (Hu et al., 2023) on CIFAR-FS for 5-way 1-shot and 5-shot learning. The results are shown in Tab. 7. ROSY brings consistent defense gains (1.48% ~ 1.55% for 1-shot learning and 1.86% ~ 2.02% for 5-shot learning) under poisoning rates of 20% and 40%, indicating its general effectiveness for existing DFML algorithms.

**Effect of the number of pre-trained models on TEAPOT and AVERAGE.** Tab. 8 and Tab. 9 show the effect of the number of pre-trained models on TEAPOT and AVERAGE, respectively. We conduct experiments on CIFAR-FS for 5-way 1-shot and 5-shot learning, respectively. Within a specific range, more pre-trained models can benefit TEAPOT but damage AVERAGE. Specifically, as shown in Tab. 8, increasing the number of models from 10 to 100 can benefit TEAPOT with $\uparrow 2.48\%$ and $\uparrow 27.28\%$ performance gains for 1-shot and 5-shot learning, respectively. However, in the case of AVERAGE, this results in $\downarrow 4.03\%$ and $\downarrow 9.88\%$ performance degradation for 1-shot and 5-shot learning, respectively. The reason why AVERAGE even performs worse than RANDOM is that each pre-trained model trains on different tasks, thus lacking precise correspondence among different pre-trained models in parameter space. The poor performance of AVERAGE also highlights the challenge of utilizing numerous available pre-trained models from sources like GitHub to enable efficient learning on unseen downstream tasks. In contrast, our proposed TEAPOT effectively solves this challenging problem and achieves significant performance improvements.

