# OpenReview forum: "Task-Distributionally Robust Data-Free Meta-Learning"
_ICLR.cc/2024/Conference — ICLR 2024 Conference Withdrawn Submission_

### Official Review · Reviewer_iMtM · 2023-10-26

**Soundness:** 2 fair
**Presentation:** 2 fair
**Contribution:** 2 fair
**Rating:** 3
**Confidence:** 4

**Summary:**

The paper studies the robustness of a meta-learning approach called data-free meta-learning (DFML), which uses only pre-trained models. The paper argues that DFML is subject to two risks: (i) task-distribution shift (TDS) and (ii) task-distribution corruption (TDC). TDS is the risk of meta-learning over-fitting due to over-reliance on the new task. TDC is the risk of contamination of pre-trained models that differ from the provided explanations (model-poisoning attack), resulting in poor meta-test performance. The paper addresses these two risks through (i) interpolated task-memory replay, which prevents overfitting by mixing past tasks with the current task, and (ii) robust model selection policy, which uses reinforcement learning to score model reliability for model selection. Experiments on representative Few-shot learning datasets show that the proposed method is useful to obtain a certain robustness for TDS and TDC. However, the risk of TDC is not practical because the model-poisoning attack can be avoided by lightweight sanity check with input data inverted from pre-trained models or generated by text-image generative models like Stable Diffusion. In my opinion, the paper should be resubmitted by discarding the claims about TDC because the risk of TDS is sufficient to discuss using the whole of the paper.

**Strengths:**

+ The paper discusses robustness for the first time in the relatively new meta-learning problem setting of DFML.
+ The paper proposes an effective method for replaying past tasks in DFML and confirms its effectiveness through experiments.
+ The paper proposes a reinforcement-learning-based model selection method that can maintain meta-testing performance even when unreliable models are included.

**Weaknesses:**

- The risk of TDC is avoidable through sanity checks with simple data collection, and thus there is little motivation to introduce the proposed method with high computational complexity using RL. In fact, the experiments provided in the paper use pre-trained models in the medical domain for the attack, but these are not difficult to determine by simply inputting data obtained from an image search engine or a text image model such as Stable Diffusion without privacy concerns. It is unnatural for the problem setting to allow data inversion for meta-learning but not to allow such a simple preprocessing. We could not agree with the practicality of TDC's problem setting unless it shows important use cases or experimental results, indicating it cannot be solved without introducing the proposed method.
- The replay-based method in TEAPOT has already been well studied in the context of continual learning [a], and there is nothing new about it other than the task (i.e., DFML) to which the method is applied.
- Section 4 is misleading because of the mixture of existing and proposed methods. In particular, the second paragraph is almost the same as in the previous study [b].
- Experiments are conducted only on 4-layered convolution neural networks, and the performance is not verified in a realistic setting with heterogeneous architectures. Since the paper considers a practical problem setting for DFML, experiments using more architectures would help make the paper more convincing.

[a] Wang, Liyuan, et al. "A comprehensive survey of continual learning: Theory, method and application." arXiv preprint arXiv:2302.00487 (2023).

[b] Hu, Zixuan, et al. "Architecture, Dataset and Model-Scale Agnostic Data-free Meta-Learning." CVPR 2023.

**Questions:**

- Can the risk of TDS be avoided by splitting the inverted dataset for validation? In the context of generative models, methods for validation and model selection using synthetic data have been proposed [c].
- Can model-poisoning attacks be detected as OOD by visualizing or embedding data recovered from a pre-trained model into an existing model? The previous study [b] actually visualized the restored data, and pre-trained neural networks can be used for OOD detection [d].
- How much does the computational complexity of ROSY reinforcement learning increase relative to the baseline (PURER)?

[c] Shoshan, Alon, et al. "Synthetic data for model selection." ICML 2023.

[d] Lee, Kimin, et al. "A simple unified framework for detecting out-of-distribution samples and adversarial attacks." NeurIPS 2018.

---

> ### Author Response · Authors · 2023-11-15
> **Response to Reviewer iMtM (Part1/2)**
>
> **Q: The risk of TDC is avoidable through sanity checks with simple data collection, by simply inputting data obtained from an image search engine or a text image model such as Stable Diffusion without privacy concerns.**
>
> A: The concern about the feasibility of sanity checks for avoiding TDC is worthy of consideration. However, our assumption in the context of Data-Free Meta-Learning (DFML) is based on scenarios where an extensive pool of pre-trained models is available. For instance, MAML, a classical meta-learning algorithm, requires over 100K tasks for effective training. Manually conducting sanity checks across such a vast array of models is impractical, both in terms of time and resources. Specifically, (i) gathering specific test data or generating specific images is time-consuming, and (ii) even if we already have the data, evaluating a large volume of models individually can be costly.
>
> In contrast, our solution allows for automatic pre-trained model identification and selection within the meta-training process, which mitigates the need for manual data collection or generation and model evaluation. This is particularly beneficial in scenarios involving constructing large model pools by automatic commands.
>
> ------
>
> **Q: The replay-based method in TEAPOT has already been well studied in the context of continual learning.**
>
> A: The reviewer's observation about the replay-based method's familiarity with continual learning is acknowledged. However, as demonstrated in our paper (Table 3 in our first submission), applying conventional replay methods to Data-Free Meta-Learning (DFML) faces unique challenges. The limited diversity in stored memory tasks does not adequately represent the underlying task distribution required for DFML, e.g. MAML requires 100K tasks for efficient training. Our proposed interpolated task-memory replay technique, as explained in Section 4.1, addresses this by generating diversified tasks from existing memory without exceeding memory constraints, thereby enhancing the meta-learner's generalization capability across a broader range of tasks. Overall, we highlight the challenges faced by existing replay-based methods in the context of data-free meta-learning and propose a simple but effective baseline.
>
> ------
>
> **Q: Section 4 is misleading because of the mixture of existing and proposed methods. In particular, the second paragraph is almost the same as in the previous study.**
>
> A:  We understand the reviewer's concern regarding the clarity of Section 4. The loss function we described, which includes a classification loss and a regularization term, is indeed a standard approach in the field of data-free learning, including data-free knowledge distillation [1,2] and model inversion [3]. Our contribution, however, is distinct from existing methods like Purer in addressing the task-distribution shift issue.
>
>  We would like to highlight the difference between the existing work Purer and ours.
>
> (i) Our work is designed to alleviate the task-distribution shift issue observed in Purer via introducing memory bank with interpolated task memory replay.
>
> (ii) Purer keeps the dataset trainable, which is adversirially optimized with the meta-learner as illustrated in the equation below.
>
> $$\min _{\boldsymbol{\theta}} \max _{\mathcal{D}} \underset{\mathcal{T} \in \mathcal{D}}{\mathbb{E}}\left[-\mathcal{L} _{\text {cls }}(\mathcal{D})+ \mathcal{L} _{\text {meta }}(\mathcal{T} ; \boldsymbol{\theta})\right]$$
>
> Unlike Purer, our work instead freezes the generated data and puts it into the memory bank,  while only optimizing a generator to generate new data. The usage of a generator is also absent in Purer.
>
>  We recognize the importance of distinguishing our work from existing literature. Thus, we will ensure to more explicitly highlight these differences in the revised manuscript.
>
> [1] Yin H, et al. Dreaming to distill: Data-free knowledge transfer via deepinversion, CVPR2020.
>
> [2] Fang G, et al. Contrastive model inversion for data-free knowledge distillation, IJCAI2021.
>
> [3] Hatamizadeh A, et al. Gradvit: Gradient inversion of vision transformers, CVPR2022.

---

> ### Author Response · Authors · 2023-11-15
> **Response to Reviewer iMtM (Part 2/2)**
>
> ------
>
> **Q: The performance is not verified in a realistic setting with heterogeneous architectures.**
> A: We acknowledge the reviewer's point on the limited scope of our experiments. To address this, we expanded our experimental setup to include diverse architectures.  The table below demonstrates the architecture-agnostic nature of our TeaPot. Here, we conduct experiments on CIFAR-FS with various model architectures.
>
> |               Model pool               | 5-way 1-shot | 5-way 5-shot |
> | :------------------------------------: | :----------: | :----------: |
> |               100% Conv4               |    40.80     |    57.11     |
> | 33% Conv4 + 33% ResNet12 +33% ResNet18 |    41.94     |    57.02     |
>
> ------
>
> **Q: Splitting Inverted Dataset for Validation to Avoid TDS.**
> A: Your comments are insightful. However, there are some issues with monitoring the training performance using a split inverted validation set.
> (i) *Compromised performance:*
>
> As shown in the table below, using 20% inverted dataset as a validation dataset to monitor the meta-training leads to a reduction in the diversity of meta-training tasks and a consequent performance drop. This is because meta-learning is evaluated on unseen tasks, which means the tasks (classes) in the validation dataset must be unseen during the meta-training. For example, if the class *dog* is used for validation, it can not be used for meta-training, which is different from traditional supervised learning.
>
> |            Validating method             | 5-way 1-shot | 5-way 5-shot |
> | :--------------------------------------: | :----------: | :----------: |
> | using 20% inverted dataset as validation |    36.18     |    52.03     |
> |              TeaPot (ours)               |    40.80     |    57.11     |
>
> (ii) *Additional time costs:*
>
> Frequent validation during meta-training adds significant time costs, as shown in the table below. In this experiment, we validate the meta-learner for every 200 meta-iterations. Our method, however, does not involve any validation during meta-training.
>
> |                  Method                  | Time (h) for 10000 meta-iterations |
> | :--------------------------------------: | :--------------------------------: |
> | using 20% inverted dataset as validation |                4.85                |
> |              TeaPot (ours)               |                2.65                |
>
> (iii) *Bias to the validation dataset:*
>
> When using an additional validation dataset, there's a risk of bias towards the validation dataset, which may not accurately predict performance on meta-testing tasks, especially due to the distribution gap between inverted validation data and real meta-testing data. Our method, however, does not involve model selection using specific criteria, focusing on maintaining high accuracy over time.
>
> ------
>
> **Q: Can model-poisoning attacks be detected as OOD by visualizing or embedding data recovered from a pre-trained model into an existing model?**
> A: In the context of our work on Data-Free Meta-Learning (DFML), where a large pool of pre-trained models is used, manually validating each model for out-of-distribution (OOD) content is impractical due to the significant time and resource requirements.
>
>  Therefore, manually validating each model might be impractical. The method you mentioned (i) is impractically time-consuming, which involves evaluating a large volume of models, and (ii) assumes the accessibility to additional "reliable" pre-trained networks.
>
> Our proposed Robust Model Selection Policy (ROSY) automatically identifies and selects reliable pre-trained models, thus streamlining the process and enhancing the efficiency of handling large model pools in DFML scenarios.
>
> ------
>
> **Q: Computational Complexity of ROSY Relative to Baselin?**
>
> A: The addition of ROSY to our TeaPot framework increases the computational time minimally, as shown in the table below. The efficiency of ROSY is due to the rapid computation of rewards and the straightforward policy optimization on reliability scores (100 scale weights), which does not involve complex gradient calculation and backpropagation.
>
> |    Method     | Time (h) for 10000 meta-iterations |
> | :-----------: | :--------------------------------: |
> |    TeaPot     |                2.65                |
> | TeaPot + RoSy |                2.70                |
>
> Moreover, our method is faster than the SOTA DFML methods, Purer. The table below shows the total time needed to achieve the best performance, which is evaluated on CIFAR-FS 5-way 1-shot learning. Compared to Purer, ours can achieve higher PEAK time while taking less time, which is also shown in Fig.3 in our first submission.
>
> |    Method     | PEAK accuracy | Time (h) |
> | :-----------: | :-----------: | :------: |
> |     Purer     |     36.26     |   2.49   |
> | TeaPot (ours) |     40.80     |   2.12   |
>
> We promise to provide a comprehensive analysis of the complexity in our revised manuscript.

---

### Official Review · Reviewer_eZp4 · 2023-10-28

**Soundness:** 2 fair
**Presentation:** 1 poor
**Contribution:** 2 fair
**Rating:** 5
**Confidence:** 4

**Summary:**

This paper proposes improved data-free meta-learning algorithms for handling task distribution shifts and adversarially injected models. TeaPot maintains a memory bank of old tasks to mitigate distribution shift, and RoSy ranks models according to a learned reliability score to prune out bad models.

**Strengths:**

The paper empirically evaluates the proposed methods and shows substantial performance and robustness gains. Table 2 considers two meta-learners (ANIL, ProtoNet) and peak and last-iteration performance.

**Weaknesses:**

The paper has major clarity issues.
- The text uses a lot of jargon and acronyms (TDS, TDC, DFML, RML, RDFML, MPA, PR...), many of which I was unfamiliar with despite actively participating in meta-learning or robustness research for years. It took a lot of work to parse on my first pass. I think the general ICLR audience would find this paper hard to understand. I'd strongly suggest a major rewrite to make the paper easier to read through, especially given that many of these concepts are simple enough to write in plain text.
- The paper proposes two methods (TeaPot and RoSy), and it's not clear how these fit together into a cohesive contribution. The paper describes TeaPot as a "DFML baseline" for handling distribution shifts over tasks (TDS) and calls RoSy a defense strategy for adversarial models (TDC).

I'm not convinced of the relevance of the model poisoning attack scenario (TDC, MPA). This assumes that a malicious attacker publicly releases a model that is a valid classifier but with the wrong task description. How realistic is this scenario, and isn't such a "poisoned model" easy to detect in practice by testing on some data that the model claims to work on? If the problem setting is that these models are actually posted online, wouldn't such "poisoned models" be quickly flagged through GitHub issues or Huggingface community discussions by someone who tried to use the model but failed? Or are there common black-box "model pools" that I'm not aware of?

(minor) Fig 4 is really hard to parse because it tries to combine two axes of variation (shot, MPA) into one. Can't you just have two 2x2 grids side-by-side, where for example the two grids are 1-shot and 5-shot?

**Questions:**

Is the paper meant to be about solving two different problems? It currently feels like two papers combined into one.

---

> ### Author Response · Authors · 2023-11-14
> **Response to Reviewer eZp4**
>
> **Q: Jargon and Acronyms Usage (TDS, TDC, DFML, RML, RDFML, MPA, PR...).**
>
> A: Thank you for pointing out the problems of readability and clarity in our manuscript. We recognize that excessive use of jargon and acronyms can hinder understanding. We will carefully revise the manuscript to minimize the use of acronyms and explain concepts in plain language wherever possible. Our focus will be on clearly defining and using essential acronyms like Data-Free Meta-Learning (DFML), Task-Distribution Shift (TDS), and Task-Distribution Corruption (TDC), ensuring that these are introduced and explained at their first occurrence and used consistently throughout the paper.
>
> ------
>
> **Q: Model Poisoning Attack Scenario (TDC, MPA) Relevance. Realism of the scenario, detectability of poisoned models, and existence of common black-box model pools.**
>
> A: Your concern regarding the detectability of poisoned models is insightful. Our assumption in the context of Data-Free Meta-Learning (DFML) is based on scenarios where an extensive pool of pre-trained models is available. Manually validating each model might be impractical. This is because (i) gathering specific test data for hundreds or thousands of models is not only time-consuming but also potentially hindered by privacy concerns, and (ii) evaluating a large volume of models individually, even with available data or public discussions, can be prohibitively expensive.
>
> The key advantage of our proposed framework is its ability to automate model selection within the meta-training process, mitigating the need for manual, model-specific data collection and evaluation. This approach is particularly beneficial in scenarios involving collecting a large number of models by automatic commands. Besides, we hope our work can inspire future works for more complicated scenarios like meta-learning from black-box API pools [1] such as Cloud Vision API of Google, Amazon AI and TensorFlow Lite APIs.
>
> [1] Zixuan Hu, et al. "Learning to Learn from APIs: Black-Box Data-Free Meta-Learning." ICML 2023.
>
> ------
>
> **Q: Is the paper meant to be about solving two different problems? It currently feels like two papers combined into one.**
>
> A: We promise that we do not combine two papers into one. We organize our paper from the general perspective of task-distributionally robustness, focus on task-distribution shift and corruption issues observed in existing methods. The shift issue is a common problem, while the corruption issue is a more specific problem arising from specific attacks. Our solutions for these two issues are not separated, which can be used together with promising results, as evidenced by the results presented in Table 4 of our initial submission.
>
> ------
>
> **Q: (minor) Fig 4 is really hard to parse because it tries to combine two axes of variation (shot, MPA) into one. Can't you just have two 2x2 grids side-by-side, where for example the two grids are 1-shot and 5-shot?**
>
> A: We appreciate your suggestion regarding Figure 4. Based on your feedback, we will redesign this figure to enhance clarity and comprehension. The revised figure will feature two separate 2x2 grids, one for 1-shot and the other for 5-shot scenarios, to distinctly present the variations in both dimensions.

---

### Official Review · Reviewer_j2rx · 2023-10-30

**Soundness:** 2 fair
**Presentation:** 3 good
**Contribution:** 3 good
**Rating:** 3
**Confidence:** 4

**Summary:**

This paper shows that current data-free meta-learning methods suffer from two drawbacks steming from constucting pseudo tasks.
In order to handle these issues, this paper proposes a memory-based DFML baseline and a defense strategy to select reliable models. The effictiveness of method was demonstrated through few-shot learning tasks.

**Strengths:**

This paper reveals the distribution of pseudo tasks will change as the learnable dataset gets updated.
Moreover it introduces model poisoning attack as a training-time attack by injecting malicious OOD models.
To handle the task-distribution shift, it introduce a task-memory bank and constructs a interpolate N-way task
representing a mixture of old tasks. Furthermore, to defense the MPA, it proposes a model selection policy
as a learnable weight vector and formulate the denfense objective as a bi-level optimization problem.

**Weaknesses:**

- The implementation details of the interpolated task-memory replay is not well demonstrated, but in the paper it plays an important role in face of TDS.
- The paper argues it is a data-free meta-learning, but to aviod the MPA, it still need an additional validataion tasks.
- To address the TDC and TDS, it essentially needs three stage training. It needs pseudo task recovery, meta-training with task-memory replay, and the policy optimization, and thus the  total training time must be long.
- The key process is pseudo tasks recovery, but the generation performence with classification loss and the regularization term is not figured out.
- The distribution of the pesudo tasks is lack of explaination, and the Figure 1 shows the degradation of the meta-testing accuracy, but it may not caused by the task-distribution shift.

**Questions:**

- Could authors figure out the distribution shift happened in the meta-training phase?
- Why does the task generator trained with such two loss function work well? Could the author give a reasonable and foundamental explaination?
- Could authors clarify how to interpolate the old tasks?
- What's the total time need to take to achieve the performence?
- Why does the fine-tune performs so bad compared with other methods?

---

> ### Author Response · Authors · 2023-11-14
> **Response to Reviewer j2rx (Part 1/2)**
>
> **Q: Details about the interpolated task-memory replay. Could authors clarify how to interpolate the old tasks?**
>
> A: We agree that additonal clarifications about this should be provided because it is central to our framework. Interpolated task-memory replay is a simple but effective technique, enabling us to construct interpolated tasks from existing memories. Specifically, given two pre-trained models with label spaces of *(lily, sky)* and *(rose, pine tree)*, a potential interpolted tasks could be *(lily, rose)*. This tenique enables meta-training on a wider range of tasks without exceeding model and memory budegts. Our experiments in Tab.3 in our first submission demonstrate the efficacy of this simple technique, yielding a 10.09% increase in LAST accuracy and a 2.72% boost in PEAK accuracy with a budget of 100 pre-trained models.
>
> Thanks for your advice, and we promise to include a more detailed explanation and refer to Table 3 in our revised version to further clarify this technique.
>
> ------
>
> **Q: The paper argues it is a data-free meta-learning, but to aviod the MPA, it still need an additional validataion tasks.**
>
> A: Indeed, our approach necessitates a minimal set of validation tasks. However, the number of these tasks (only two in our case) is substantially fewer than the extensive task sets typically required for meta-training in traditional methods (e.g., over 100K tasks for MAML). Our method demonstrates that using a small number of validation tasks is not only practical but also incurs minimal time costs for calculating the reward in policy optimization.
>
> ------
>
> **Q: To address the TDC and TDS, it essentially needs three stage training. It needs pseudo task recovery, meta-training with task-memory replay, and the policy optimization, and thus the total training time must be long. What's the total time need to take to achieve the performence?**
>
> A: Indeed, our methods needs three stage training. However,  as illustrated in the table below, integrating policy optimization adds only an additional 0.05 hours for CIFAR-FS 5-way 1-shot learning. This efficiency is due to the rapid computation of rewards (accuracy on a small number validation tasks) and the straightforward policy optimization on reliability scores (100 scale weights), which does not involve complex gradient calculation and backpropagation.
>
> |    Method     | Time (h) for 10000 meta-iterations |
> | :-----------: | :--------------------------------: |
> |    TeaPot     |                2.65                |
> | TeaPot + RoSy |                2.70                |
>
> Moreover, our method is faster than the SOTA DFML methods. The table below shows the total time needed to achieve the best performance, which is evaluated on CIFAR-FS 5-way 1-shot learning. Compared to Purer, ours can achieve higher PEAK time while taking less time, which is also shown in Fig.3 in our first submission.
>
> |    Method     | PEAK accuracy | Time (h) |
> | :-----------: | :-----------: | :------: |
> |     Purer     |     36.26     |   2.49   |
> | TeaPot (ours) |     40.80     |   2.12   |
>
> We promise to provide a comprehensive analysis of the complexity in our final manuscript.
>
> ------
>
> **Q: The key process is pseudo tasks recovery, but the generation performence with classification loss and the regularization term is not figured out. Why does the task generator trained with such two loss function work well? Could the author give a reasonable and foundamental explaination?**
>
> A: We acknowledge the need for a more detailed explanation of the pseudo task recovery process.
>
> The classification loss in our approach ensures that the recovered images capture class-specific discriminative features, facilitating accurate classification. Concurrently, the regularization term enhances the realism of these pseudo images by aligning the statistics of their layer-wise feature maps with those of real images. We will provide visualizations of recovered images (w/ and w/o the regularization term) and elaborate on this process in our revised manuscript.
>
> ------
>
> **Q: Why does the fine-tune performs so bad compared with other methods?**
>
> A: Fine-tuning with extremely few-shot examples (1-shot and 5-shot in our cases) can result in severe overfitting, leading to poor performance on query examples. Our methods, utilizing various meta-learning algorithms, effectively facilitates efficient learning of new tasks, by meta-learning a sensitive model initialization (our TeaPot-ANIL) or a task-shared feature extractor (our TeaPot-ProtoNet).
>
> ------

---

> ### Author Response · Authors · 2023-11-14
> **Response to Reviewer j2rx (Part 2/2)**
>
> ------
>
> **Q: The distribution of the pesudo tasks is lack of explaination, and the Figure 1 shows the degradation of the meta-testing accuracy, but it may not caused by the task-distribution shift. Could authors figure out the distribution shift happened in the meta-training phase?**
>
> A: We plan to expand our discussion on task-distribution shifts in the final version of our manuscript. Previous SOTA work, Purer, uses a learnable pseudo dataset that evolves alongside the meta-learner. This evolution can lead to significant shifts in task distribution, particularly under adversarial training conditions.
>
> The overall objective of Purer is shown in the equation below.
>
> $$\min _{\boldsymbol{\theta}} \max _{\mathcal{D}} \underset{\mathcal{T} \in \mathcal{D}}{\mathbb{E}}\left[-\mathcal{L} _{\text {cls }}(\mathcal{D})+ \mathcal{L} _{\text {meta }}(\mathcal{T} ; \boldsymbol{\theta})\right]$$
>
> The learnable dataset is optimized with two terms: (i) maximizing the classification loss $-\mathcal{L} _{\text {cls }}$ensures the images inside the pseudo dataset can be correctly classified, and (ii) minimizing $\mathcal{L} _{\text {meta }}$ ensures the tasks constructed from the pseudo dataset will become harder. Therefore, as the learnable dataset gets updated, the distribution of pseudo tasks will change. Such distribution shift could be large when the learnable dataset and the meta-learner get trained adversarially.
>
> We will elucidate this with detailed analysis and illustrations in our revised manuscript to clarify how such shifts impact the meta-training phase and overall performance.

---

### Official Review · Reviewer_ywQB · 2023-11-01

**Soundness:** 3 good
**Presentation:** 4 excellent
**Contribution:** 3 good
**Rating:** 8
**Confidence:** 3

**Summary:**

Data-Free Meta-learning (DFML) focuses on quickly learning new tasks without using original training data by leveraging a collection of pre-trained models. Traditional DFML approaches create pseudo tasks from a learnable dataset that is derived from these pre-trained models. This method has two main issues: Task-Distribution Shift (TDS) and Task-Distribution Corruption (TDC). TDS arises when the pseudo task distribution changes as the learnable dataset evolves. TDC, on the other hand, occurs when deceptive models are deliberately added to the collection. To counter these problems, the paper introduces a robust DFML strategy. Firstly, authors present TEAPOT, a memory-based DFML solution to manage TDS by keeping older task memories and diversifying the pseudo task distribution. Secondly, to defend against TDC, they suggest a Robust Model Selection Policy (ROSY) that assesses model reliability and is compatible with current DFML methods.

**Strengths:**

1. The paper adeptly identifies and addresses the significant challenges of task-distribution shift (TDS) and task-distribution corruption (TDC) within data-free meta-learning.

2. The paper astutely recognizes the prevailing issue of task-distribution shift, particularly spotlighting the shortcomings in prior research like PURER. As a remedy, the authors introduce a novel method that maintains a memory of previously generated distribution samples.

3. The introduction of the task-distribution corruption problem is a pioneering effort in the data-free meta-learning context. The provided examples and rationale eloquently underscore its importance and the pressing need for an effective solution.

4. The empirical evidence presented is thorough, conclusively demonstrating the superiority of the proposed method under the aforementioned challenges.

**Weaknesses:**

The benchmarking approach against the prior SOTA (PURER), which utilizes DeepInversion for data synthesis, raises questions. In the proposed method, data specific to the task is synthesized using a generator. It remains ambiguous, particularly in Table 2, whether the authors employed consistent sample-generating strategies for both PURER and their proposed method. For a more equitable comparison, it would be beneficial if the authors utilized a uniform sample-generating approach, be it DeepInversion, the generative model, or both, for both PURER and their method.

**Questions:**

See weakness